# The Role of Liquidity Creation in Managing the COVID-19 Banking Crisis in Selected Mena Countries

**Hani El-Chaarani [1], Rebecca Abraham [2,\*] and Georges Azzi [3]**

[1]  Faculty of Business Administration, Beirut Arab University, Tripoli Campus, Riad El Solh, Beirut P.O. Box 1150-20, Lebanon

[2]  Huizenga College of Business, Nova Southeastern University, 3301 College Ave, Fort Lauderdale, FL 33314, USA

[3]  Business School, Holy Spirit University of Kaslik, Jounieh P.O. Box 446, Lebanon

\*  Correspondence: abraham@nova.edu

**Abstract:** Banks are financial intermediaries who transform deposits into loans. Banks in the MENA (Middle East and North Africa) region use large deposits from oil companies and big businesses to finance trade, and fund government and private sector infrastructure projects. The role of banks in financing trade and development is significant as undeveloped capital markets are unable to perform this function. During the COVID-19 crisis, banks sustained liquidity shocks, as deposits were withdrawn to meet personal and business needs. Essentially, banks could not make loans, as the funds to make loans were depleted. The purpose of this study is to evaluate the effectiveness of liquidity creation as a main force, in conjunction with other performance predictors such as efficient asset management, asset quality, and bank size, on bank financial performance, either individually or in conjunction with liquidity creation during the COVID-19 financial crisis. We used bank financial data from a sample of 298 banks from 11 countries in the MENA region, including Egypt, Tunisia, Morocco, Qatar, Bahrain, Oman, Kuwait, Saudi Arabia, United Arab Emirates, Jordan, and Israel, from 2020 to 2021. Liquidity creation, efficient asset management, asset quality, and bank size increased bank return on assets and return on equity. Bank size and asset quality acted jointly with liquidity creation to increase return on assets and increase return on equity. We conclude that as liquidity creation acts individually, and in conjunction with asset quality and bank size to increase bank profits, both its main effect and its moderated effect, can maintain bank profitability, during periods of extreme liquidity supply shocks, such as the COVID-19 crisis.

**Keywords:** liquidity; MENA; banking crises; bank financial performance; asset quality; bank size

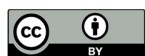

## 1. Introduction

Banking dominates as the chief source of corporate funding and a significant source of government funding in the MENA (Middle East and North Africa) region (Mdaghri 2022). Banks hold assets worth up to 60–100% of GDP across MENA countries (Ghosh 2017). The basic function of a bank is to be a financial intermediary that transforms deposits to loans.

This function can only be performed if there are sufficient deposits. The accumulation of deposits has been described as a net defensive position, against liquidity risks, as the funds are used to fund liquidity scarcities (Kesraoui et al. 2022). During a financial crisis, such as the COVID-19 lockdown, both governments and business depositors withdrew their bank deposits to fund their own emergency financial needs. Individuals facing unemployment drained their deposits as well. The shrinking level of deposits inhibited the flow of loans from banks to businesses and government. This was a liquidity shock.

Banks responded by introducing strategies to increase the flow of deposits. In the first strategy, El-Chaarani et al. (2022) describe the importance of large shareholders, who

were also politicians. They used their political connections to obtain preference in government funding and some private-sector funding. The flow of deposits was maintained, or liquidity creation ensued, albeit at less than pre-pandemic levels. Independent directors on the board of directors made liquidity creation a criterion for the performance of senior managers. In this study, we determine if such liquidity creation was effective in increasing bank profits during the COVID-19 crisis. We found that academicians could add liquidity creation and other predictors to the list of variables that assist banks in achieving profitability during a crisis. Practitioners could then employ these corporate governance measures to increase liquidity creation, and maintain bank profitability, during a financial crisis. Therefore, this study contributes to our knowledge of the financial variables that assist banks in remaining profitable during periods of severe liquidity shocks, i.e., when deposit drains are frequent.

The second strategy to maintain bank profitability during the COVID-19 crisis was to manage assets efficiently. Efficient asset management consisted of converting assets into operating income. Operating assets are primarily loans, and unused loan commitments. By offering attractive credit terms, and a variety of payment options, certain banks were able to retain borrowers, thereby sustaining loan income. Unused loan commitments provided a capital reserve to be employed in the creation of new loans. Lei et al. (2021) observed that unused loan commitments provided a significant source of funds for US banks during the COVID-19 banking crisis. The third strategy was to improve asset quality by reducing the volume of nonperforming loans. Intuitively, more stringent credit assessment resulted in the rejection of new loan applications from less creditworthy businesses and individuals. Offering incentives to speed payment may have stimulated certain borrowers to resume payment on nonperforming loans. In an examination of Bangladeshi banks during the COVID-19 lockdown, Rahman et al. (2021) found that the reduction of nonperforming loans in a capped interest rate environment significantly increased bank liquidity.

Finally, bank size may have permitted large banks to cope more effectively with both supply and demand shocks to financial intermediation. Large banks have a variety of income streams, so that deposit drains may be offset by alternative sources of revenue. Loan commitments, derivatives, letters of credit, jumbo certificates of deposit, lending federal funds to other banks, and making repurchase agreements are some of the products sold by large banks that are inaccessible to smaller institutions. Large banks also have a diverse array of borrowers. Individual and commercial mortgage holders, large- and medium-sized business loans, credit cards with a range of maturities, and automobile loans with varying interest rates for risky and less risky borrowers are some of the borrowers who create multiple revenue streams for large banks that are not available to smaller financial institutions. Flynn and Wu (2022) observed that bank size was a significant explanatory variable of the variation in bank lending at US banks during the COVID-19 banking crisis.

Academic literature on liquidity creation in the MENA region has been primarily confined to the pre-2017 period, suggesting the need for current literature. Mohammed (2014) examined liquidity creation in banks in 18 MENA countries, finding that large banks created more liquidity than small banks and SME banks. Liquidity creation had a positive impact on return on assets for large banks, and an adverse effect on return on assets for their small- and medium-sized counterparts. These results were partially supported by Sahyouni and Wang (2019), whose evaluation of MENA banks in the 2011–2016 period found a significant negative correlation between liquidity creation and return on equity. In contrast, Mdaghri (2022) found that liquidity creation among MENA banks from 2010 to2017, diminished the level of nonperforming loans, suggesting that bank liquidity creation served an economic enhancement function. In short, banks in the pre-2017 period were advised to increase capital and liquidity creation, if they were large in size.

Literature on liquidity creation during the COVID-19 crisis in the MENA region is sparse. El-Chaarani et al. (2022) focused on the effects of corporate governance on bank

performance, which facilitates liquidity creation, but does not measure it. El-Chaarani et al. (2022) were concerned with the power relationships between politicians and managers, and their impact on bank profitability. There was no mention of the role of bank liquidity creation in influencing bank profits. Kesraoui et al. (2022) addressed liquidity risk, finding that MENA banks increased interest margins to account for liquidity risk during downturns. However, their study period of 2004–2020 included just one year of the COVID-19 crisis, so it cannot be considered representative of the lockdown period. Mdaghri (2022) and Sahyouni and Wang (2019) recognized the importance of liquidity creation during a financial downturn, but covered the pre-lockdown, 2010–2017 period. They cautioned against excessive liquidity creation, as the creation of more loans could result in a larger volume of nonperforming loans. The literature on the impact of COVID-19 lockdowns on bank liquidity is largely confined to developed economies in Western countries (see Lei et al. (2021) and Flynn and Wu (2022)), so that this study of bank liquidity in the MENA region contributes to knowledge of banking during crises in a different location.

The COVID-19 pandemic of 2020–2021 was a black swan event. It resulted in unprecedented lockdowns of individuals and businesses, disrupting the flow of funds from depositors to lenders to borrowers. Worldwide, the scale of financing was found to have decreased (Lei et al. 2021), along with poor financial performance (Rizwan et al. 2020) and increased operating risk. Simultaneously, banks experienced funding demand shocks as borrowers in business responded to declining business revenues by defaulting on existing loans. Defaults on different types of loans reduced bank liquidity (Sivaprasad and Matthew 2021). It is of academic importance to empirically validate the methods used by banks to maintain profitability, to add to theoretical knowledge of sources of bank profitability during crises. In the MENA region, there were large banks with significant reserves to maintain deposits for liquidity creation, particularly in the Gulf Cooperation Council (GCC) countries. There are smaller institutions as well, that meet the liquidity needs of SMEs and individuals. It follows that the MENA region, with its varying levels of oil wealth and development, provides a fertile ground within which to examine disruptions to banks that vary in size and liquidity.

There is no comprehensive examination of the effectiveness of the aforementioned coping strategies, or bank size, in the MENA region during the 2020–2021 COVID-19 banking crisis. To close this research gap, we undertook a two-fold examination of the predictors of profitability and effect of loan expansion across banks in the 11 MENA countries of Egypt, Tunisia, Morocco, Qatar, Bahrain, Oman, Kuwait, Saudi Arabia, United Arab Emirates, Jordan, and Israel. First, we explored the determinants of financial performance of MENA banks to identify the financial variables, such as capitalization, size, asset management, and quality of assets, that explained the profitability measures of return on assets and return on equity. Then, we examined the impact of liquidity. Liquidity was defined as loans/assets, or the quantity of loans, scaled by asset size. We tested the moderating influence of liquidity on the aforementioned financial variables, as we conjectured that increased liquidity would increase the positive influence of capitalization, size, and quality of assets on return on assets and return on equity.

The remainder of this paper is organized as follows. Section 2 is a literature review. Section 3 justifies the hypotheses. Section 4 consists of results. Section 5 describes the methodology, while Section 6 lists the conclusions.

## 2. Literature Review

The previous section identified capital requirements, bank size, reduction of nonperforming loans, and liquidity creation as predictors of bank financial performance in the MENA region. Therefore, our literature review consists of (1) the role of liquidity creation in banking, (2) liquidity creation in the MENA region, and (3) predictors of bank performance in the MENA region before and during the COVID-19 periods.

*2.1. The Role of Liquidity Creation in Banking*

Banks are liquidity creators. They provide liquidity to both depositors and borrowers. To depositors, they provide interest income in the form of a liquid cash stream. The risk-enhancing entities, by providing loans to borrowers, bestow liquidity upon borrowers, while assuming illiquidity risk. For example, upon granting a mortgage, the bank obtains an illiquid asset of the mortgage, while the borrower's responsibility is limited to a stream of liquid payments (Berger and Bowman 2016). Yet, the liquidity creation, which is the subject of this study, is the transformation of bank deposits into loans.

Liquidity creation furthers the profit-maximizing goals of banks. The seminal Klein-Monti model posited that banks earn a profit by maximizing the spread between loans and deposit rates (Klein 1971; Monti 1972). A robust supply of deposits ensures the cash inflow to create loans. During stable economic periods, the cash inflow remains at a constant level. During an economic downturn, such as during the COVID-19 crisis, deposit drains ensue, with depositors withdrawing a part or all of their deposits. Liquidity risk occurs as the supply of loans diminishes for lack of deposits. Dermine (1986) describes this phenomenon as a profit-lowering cost, as banks simply make fewer loans to businesses and individuals, with an adverse impact on profits. Alternatively, liquidity risk increases bank interest margins, as banks increase interest rates to maintain profits at the pre-downturn levels. The few loans available carry higher interest rates. Kesraoui et al. (2022) observed the increase in bank interest margins due to liquidity risk for a sample of MENA banks from 2004 to 2020. Ayedmir and Guloglu (2017) supported this finding for a Turkish sample that showed an increase in bank spreads during economic downturns.

### 2.2. Predictors of Bank Financial Performance in the MENA Region in the PreCOVID-19 Era

One of the earliest datasets of MENA banks was employed by Bitar et al. (2016), covering the 1999–2013 period. They found that banks with high Tier 1 (equity) and Tier 2 (long-term debt and equity) capital ratios had higher loan loss reserves and higher profitability. These results were particularly important for large banks, banks in countries with crises, and banks in environments with good governance. Such banks may be more prudent in lending, due to access to more information about the creditworthiness of borrowers for large banks, economic pressure for banks in crisis, and board oversight for banks in countries with strong governance. Larger capital reserves and professional management gave banks in Gulf Cooperation Council countries an advantage over their counterparts in the rest of the MENA region.

Sahyouni and Wang (2019) compared the performance of Islamic and conventional banks from 2011 to2016. They defined liquidity creation as the conversion of liquid assets to illiquid liabilities (swap treasury bills in exchange for long-term debt), or financing of illiquid assets (long-term loans), with liquid liabilities (short-term deposits). Loan commitments may also be used to create liquidity. Islamic banks were more effective than conventional banks in creating liquidity. Liquidity creation had mixed effects on profitability, in that it reduced return on average equity while having no effect on return on average assets.

Mdaghri (2022) examined the impact of liquidity creation on non-performing loans. He observed that liquidity creation through the creation of liquid assets, the financing of illiquid assets, and loan commitments, reduced the level of nonperforming loans in both Islamic and conventional banks. The reduction in nonperforming loans was theorized to have a positive influence on financial performance, with the loss of income from nonperforming loans being replaced by increased income from performing loans. Such income may be used to finance economic growth and development. Fidrmuc et al. (2015) observed the positive impact of such liquidity creation by Russian banks on economic growth. Jabbouri and Naili (2019) observed improved bank debt servicing during periods of high economic growth, as borrowers accelerated debt repayments.

Hakimi et al. (2022) evaluated liquidity risk among MENA banks during the 2004–2015 period. An increase in liquidity risk significantly decreased the profitability of MENA banks in terms of reduced return on assets and reduced return on equity. They

attributed this finding to the maturity transformational role of banks. Banks have illiquid long-maturity assets and liquid short-term liabilities. Loan defaults during crisis periods raised concerns among depositors that the income from long-term assets would not be available to fund their deposits, leading to deposit withdrawals.

*2.3. Predictors of Bank Financial Performance in the MENA Region in the COVID-19 Period*

There are few scholarly articles on measures of bank profitability in the MENA region. Two recent studies measured the effectiveness of corporate governance mechanisms on bank profitability. The basic purpose of corporate governance lies in regulating management's actions through a control and monitoring system that upholds shareholder wealth maximization. Therefore, El-Chaarani et al. (2022) hypothesized that banks that implemented these measures could be expected to increase return on assets and return on equity. They evaluated 158 banks in the 12 MENA countries of Qatar, Oman, Bahrain, Saudi Arabia, Egypt, Kuwait, Jordan, Morocco, United Arab Emirates, Tunisia, Israel, and Iran. Results indicated that the presence of independent directors on the board, several large shareholders on the board, lack of political pressure on board members, and an environment with strong legal protections increased bank return on assets and return on equity. El-Chaarani and Abraham (2022) found similar results in an examination of 194 Lebanese banks during the 2020–2021 banking crisis in Lebanon. The presence of independent directors and large shareholders due to family ownership increased profits as they had in the aforementioned study of selected MENA countries. In addition, these corporate governance measures increased liquidity levels and liquidity creation. Increased scrutiny of management by the board, and activism by large shareholders, led to the implementation of measures that increased the inflow of bank deposits, and in turn, loan creation. Management became more aligned with shareholder wealth maximization, so that they pursued strategies to attract depositors by offering incentives, preferred interest rates, and special cash management accounts. Such actions reduced fund withdrawals, facilitating loan creation.

## 3. Hypotheses Development

### 3.1. The Conceptual Model

The hypothesized relationships in this study are as follows.

- **Direct Relationships**

    Liquidity creation → return on assets, return on equity
    Equity capital → return on assets, return on equity
    Management of assets → return on assets, return on equity
    Asset quality → return on assets, return on equity
    Bank size → return on assets, return on equity

- **Moderated Relationships**

    Liquidity creation*bank equity capital → return on assets, return on equity
    Liquidity creation*bank size → return on assets, return on equity
    Liquidity creation*management of assets → return on assets, return on equity
    Liquidity creation*asset quality → return on assets, return on equity

### 3.2. Equity Capital and Profitability

While earlier theories, such as the pecking order theory and the tradeoff theory, placed less emphasis on the use of equity capital due to its high costs, later theories, such as market timing and income anticipation, envisioned a role for equity financing of banks. During crises, the investment in bank equity capital by affluent residents of MENA countries is justified, as debt financing becomes too risky for investors. This is the displacement of risky debt due to market conditions described by Baker and Wurgler (2022). The income anticipation theory (Holmstrom and Tirole 1997) also views equity capital as a source of

bank financing in the event that market conditions curb lending. Such market conditions occurred during the COVID-19 lockdown. Investors may have viewed debt as increasing bankruptcy costs, with defaults on long-term loans issued by banks. The overvaluation of debt increased investment in less risky equity capital.

In accordance with Bitar et al. (2016), we suggest that the increase of Tier 1 capital, consisting of retained earnings, results in reinvestment of bank profits to stimulate growth and profitability. This strategy is recommended in the MENA region, as MENA banks were found to have higher Tier 2 capital ratios than Tier 1 capital ratios.

**Hypothesis 1.** *Investment in bank equity capital increased bank return on assets and bank return on equity.*

### 3.3. Management of Assets

Asset management is the ability to manage and maximize the operating income of a financial institution, by increasing income and lowering costs. For example, banks derive revenue from loan commitments and letters of credit. They can reduce loan processing expenses by selling these assets in bulk, or using technology to automate the loan processing process. Liu and Wilson (2010) set forth that techniques to boost portfolio quality significantly increased bank income. Specifically, bank purchases of investment grade bonds and government securities provided income stability during periods of market uncertainty. Reviewing defaulting loans to develop creative payment alternatives may succeed in converting these loans to performing loans.

Empirical support for the ability of efficient asset management to increase bank financial performance has been provided for both MENA banks and non-MENA banks. In the MENA region, successive studies by El-Chaarani and Shaker (2018), Almaqtari et al. (2019), and Al-Homaidi et al. (2018) observed the enhancement of bank performance by efficient asset management in the pre-COVID-19 period. Lelissa (2014) and the aforementioned Liu and Wilson (2010) studies covered a similar period in non-MENA locations.

**Hypothesis 2.** *Efficient management of assets increased bank return on assets and bank return on equity.*

### 3.4. Liquidity Creation

Berger and Bowman (2016) described the financial intermediation role of banks as the creation of liquidity by transforming liquid deposits into liquid loans, while assuming illiquidity risk. Therefore, we define liquidity creation as the total loans/total assets. Banks that issue more loans are considered to create liquidity. Liquidity creation is particularly important during periods of crisis, such as the COVID-19 financial crisis, when supply shocks reduce the inflow of equity capital, and demand shocks increase non-performing loans due to defaults. In addition, large deposit withdrawals may occur. Maintaining liquidity is essential in the MENA region, as the literature has found that maintaining liquidity levels improved bank financial performance (El-Chaarani 2019) and reduced solvency risk (Al-Harbi 2019).

The benefits of liquidity creation may strengthen the positive impact of bank size, the efficient management of assets, equity capital, and asset quality on bank financial performance during crises. Large banks have large reserves to weather economic downturns. If they are able to make large loans (increase liquidity), their access to large amounts of capital will facilitate the making of loans, increasing loan interest income, and in turn, increasing profitability. If bank management can increase the conversion of assets to operating income, the availability of more loans will further increase operating income. In an examination of mergers among European banks, Varmaz and Laibner (2016) observed that profitable target banks of large size experienced significant abnormal returns upon merger announcement. If more equity capital flows into banks, in an environment of increased loan creation, more loans will be sold, increasing bank revenue. Al-Matari (2021) provided

some empirical support for this thesis, in that liquidity creation moderated the capital adequacy–profitability relationship for part of the MENA region, i.e., the Gulf Cooperation Council countries. If assets are managed efficiently, banks may impress depositors, who will increase the size of deposits, facilitating loan creation. Hassan and Giouvris (2020) observed that banks with higher capital to total asset ratios (measure of efficient management of assets) had higher loan-to-deposit ratios. If asset quality is improved by reducing nonperforming loans, and liquidity creation increases the number of new loans, new performing loans will be sold, increasing bank profitability. We may then surmise that liquidity creation moderates the relationship between bank size, asset management, capitalization, and financial performance.

**Hypothesis 3.** *Liquidity creation increases bank return on assets and bank return on equity.*

**Hypothesis 4a.** *Liquidity creation moderates the relationship between bank size, return on assets and return on equity.*

**Hypothesis 4b.** *Liquidity creation moderates the relationship between management of bank assets, return on assets and return on equity.*

**Hypothesis 4c.** *Liquidity creation moderates the relationship between bank equity capital, return on assets and return on equity.*

**Hypothesis 4d.** *Liquidity creation moderates the relationship between bank asset quality, return on assets and return on equity.*

### 3.5. Asset Quality

Asset quality is measured by the proportion of nonperforming loans. Defaulting loans are characterized by nonpayment for 90 days. During a crisis, the volume of defaults increases, leading to trepidation among depositors and investors (El-Chaarani 2019). Bloem and Gorter (2001) showed that depositors lose confidence in financial institutions with high levels of nonperforming loans, leading to withdrawals of savings. High levels of nonperforming loans have been observed to adversely impact bank profitability during the pre-COVID-19 period in the MENA region (Kadioglu et al. 2017; Saif-Alyousf et al. 2017). We wished to supplement the above studies by measuring the impact of nonperforming loans on bank profitability during the COVID-19 period. Therefore, we tested the additional moderation of asset quality on profitability by liquidity creation.

**Hypothesis 5.** *Asset quality increases bank return on assets and bank return on equity. Specifically, the increase in nonperforming loans reduces bank return on assets and bank return on equity.*

### 3.6. Bank Size and Profitability

Multiple studies have found a positive relationship between bank size and profitability in the pre-COVID-19 period in the MENA region (Al-Harbi 2019; Ali and Puah 2019; Zolkifkli et al. 2019). Large banks have financial and operational tools to generate economies of scale during crises (Adusei 2015). Investment in technology and portfolio diversification are additional sources of revenue during crises (Mirzaei et al. 2013). During the COVID-19 period, El-Chaarani et al. (2022) found that investments in fintech, big data, and technology were effective in increasing both individual and corporate customer satisfaction, resulting in a positive impact of bank size on returns. However, the study did not address the joint effect of liquidity creation and bank size on profitability, which was undertaken in this study.

**Hypothesis 6.** *Large size increases bank return on assets and bank return on equity. Specifically, the banks with large values of total assets have higher return on assets and return on equity.*

## 4. Results

### 4.1. Descriptive Statistics

Table 1 shows the means and standard deviations of dependent and independent variables of banks in the MENA region, namely Egypt, Tunisia, Morocco, Qatar, Saudi Arabia, Oman, Bahrain, Kuwait and United Arab Emirates. Qatari and Bahrani banks had higher profitability (ROA and ROE) than their counterparts in the rest of the region. Omani and Qatari banks were characterized by the lower level of financial liquidity. Banks in Bahrain, Oman, Saudi Arabia, and the United Arab Emirates had the highest levels of equity capital. The highest nonperforming loan ratio occurred in Bahrain, Saudi Arabia, and United Arab Emirates.

**Table 1.** Descriptive statistics (means and standard deviations).

| Variable/Country Mean (SD) | ROA Mean (SD) | ROE Mean (SD) | Liquidity Mean (SD) | Size Mean (SD) | Equity Capital Mean (SD) | Asset Management Efficiency Mean (SD) | Asset Quality Mean (SD) |
|---|---|---|---|---|---|---|---|
| Egypt | 0.0058 (1.1724) | 1.6422 (12.9449) | 63.338 (9.5333) | 16.3938 (5.3053) | 16.731 (11.9492) | 0.0215 (0.184) | 6.4843 (7.9581) |
| Tunisia | 0.0045 (1.0294) | 0.6742 (11.3886) | 58.484 (7.5952) | 15.9963 (1.2201) | 15.5504 (8.9209) | 0.194 (0.166) | 5.3331 (8.9941) |
| Morocco | 0.0036 (0.0585) | 0.5285 (10.4842) | 55.3553 (9.5995) | 14.5501 (1.3016) | 16.0383 (8.4909) | 0.115 (0.110) | 6.9331 (9.0941) |
| Qatar | 0.9857 (1.1589) | 6.0392 (9.0015) | 65.0392 (8.0017) | 16.9837 (1.2831) | 13.5951 (9.2010) | 0.00673 (0.193) | 4.3812 (8.4950) |
| Bahrain | 0.2421 (1.0974) | 2.3542 (12.341) | 43.2893 (8.493) | 16.4935 (1.4402) | 16.7328 (8.4232) | 0.0324 (0.165) | 7.9886 (6.5643) |
| Oman | 0.6237 (1.3034) | 3.9732 (13.449) | 68.7756 (8.0021) | 15.4945 (1.4949) | 16.7161 (8.4752) | 0.00469 (0.133) | 4.4842 (7.9390) |
| Kuwait | 0.0078 (0.9585) | 0.6645 (11.2932) | 55.7665 (9.9010) | 16.9871 (1.4941) | 12.3603 (9.4948) | 0.1320 (0.101) | 5.3932 (6.0032) |
| Saudi Arabia | 0.0641 (1.6432) | 1.5432 (14.1320) | 54.6320 (9.3097) | 1.3216 (1.5423) | 16.7266 (14.6429) | 0.0248 (0.144) | 7.0043 (7.6450) |
| United Arab Emirates | 0.0054 (1.5242) | 2.5102 (11.6354) | 54.528 (8.732) | 17.4328 (1.7765) | 16.7543 (13.7654) | 0.0253 (0.152) | 7.0143 (2.4432) |
| Jordan | 0.0074 (1.2045) | 0.6635 (12.4905) | 60.723 (8.1103) | 16.3493 (1.1121) | 12.4803 (11.5951) | 0.0145 (0.103) | 6.4904 (8.2995) |
| Israel | 0.0046 (0.8272) | 6.3946 (9.3045) | 53.7569 (8.9904) | 17.0934 (1.4453) | 16.4754 (8.1054) | 0.0453 (0.1009) | 4.7602 (6.0291) |
| Total | 0.1777 (1.1797) | 2.4534 (11.6786) | 57.6078 (8.7517) | 16.4633 (1.4014) | 15.4970 (10.2785) | 0.0293 (0.1411) | 6.0243 (7.6786) |

SD = Standard deviation.

### 4.2. Hypothesis Tests

Table 2 shows the results of the ordinary least squares regressions. Two models are presented, with Model 1 showing independent variables as the predictors and Model 2 showing the moderation by liquidity creation. Hypothesis 1 was rejected. Equity capital had no significant effect on return on equity (t = 1.3766, $p > 0.05$, Model 1), and no significant effect on return on assets (t = 1.5532, $p > 0.05$, Model 1). Hypothesis 2 was supported, as the efficient management of assets significantly increased return on equity (t = 7.136, $p < 0.001$, Model 1), and significantly increased return on assets (t = 6.919, $p < 0.001$).

Hypothesis 3 was partly supported contrary to the hypothesized direction, as liquidity creation significantly decreased return on equity (t = −9.4565, *p* < 0.001, Model 1), and significantly increased return on assets (t = −18.538, *p* < 0.001). Hypothesis 4a was supported contrary to the hypothesized direction, as liquidity creation significantly decreased the positive impact of size on return on equity (t = −8.436, *p* < 0.001, Model 2), and significantly decreased the positive impact of size on return on assets (t = −9.1530, *p* < 0.001, Model 2). Hypothesis 4b was rejected, as liquidity creation did not influence the impact of the efficient management of assets on return on equity (t = −1.545, *p* > 0.05, Model 2) and return on assets (t = −1.327, *p* < 0.05, Model 2). Hypothesis 4c was rejected, as liquidity creation did not influence the impact of equity capital on return on equity (t = −1.7653, *p* > 0.05, Model 2) and return on assets (t = −1.9255, *p* < 0.05, Model 2). Hypothesis 4d was supported contrary to the hypothesized direction, as liquidity creation significantly decreased the positive impact of asset quality on return on equity (t = −16.5293, *p* < 0.001, Model 2), and significantly decreased the positive impact of asset quality on return on assets (t = −10.5434, *p* < 0.001, Model 2). Hypothesis 5 was supported, as size significantly increased return on equity (t = 7.6094, *p* < 0.001, Model 1), and significantly increased return on assets (t = 7.8583, *p* < 0.001, Model 1). Hypothesis 6 was supported partly contrary to the hypothesized direction, as asset quality significantly decreased return on equity (t = −3.7062, *p* < 0.001, Model 1), and significantly increased return on assets (t = −3.5049, *p* < 0.001).

**Table 2.** Results of ordinary least squares regressions of return on equity and return on assets on main and moderated independent variables.

| Variable | Model 1 | | Model 2 | | Hypothesis Supported/Rejected |
|---|---|---|---|---|---|
| | ROE | ROA | ROE | ROA | |
| Constant | 1.76 *** | 1.65 *** | 1.63 *** | 1.74 *** | |
| Liquidity | −0.20 *** | −0.24 ** | −0.25 *** | −0.26 *** | Hypothesis 3 supported |
| Size | 0.08 *** | 0.094 *** | 0.09 *** | 0.09 *** | Hypothesis 5 supported |
| Equity Capital | 0.28 | 0.230 | 0.24 | 0.23 | Hypothesis 1 rejected |
| Efficient Management of Assets | 0.98 *** | 1.09 *** | 1.09 *** | 1.11 *** | Hypothesis 2 supported |
| Asset Quality | −0.13 * | −0.16 *** | −0.15 ** | −0.16 ** | Hypothesis 6 supported |
| Size*Liquidity | | | −0.43 *** | −0.41 ** | Hypothesis 4a supported |
| Equity Capital*Liquidity Creation | | | −0.30 | −0.32 | Hypothesis 4c rejected |
| Efficient Management of Assets*Liquidity Creation | | | −1.40 | −1.16 | Hypothesis 4b rejected |
| Asset Quality*Liquidity Creation | | | −2.30 *** | −2.54 *** | Hypothesis 4d supported |
| Inflation | 0.05 * | 0.06 * | 0.05 | 0.06 | |
| Gross Domestic Product | 0.10 | 0.10 | 0.12 | 0.14 | |
| R² | 0.5247 | 0.5612 | 0.582 | 0.539 | |
| N | 298 | 298 | 298 | 298 | |

* *p* < 0.05, ** *p* < 0.01, *** *p* < 0.001; ROE = return on equity, ROA = return on assets,.

### 4.3. Robustness Checks

Both generalized method of moments (henceforth, GMM), and two-stage least squares (henceforth, 2SLS) models were employed to eliminate any bias due to heteroscedasticity, endogeneity of independent variables, and the presence of unobserved independent variables. According to prior literature (see El-Chaarani et al. 2022 for a review), these models overcome such problems in banking studies. The results in Table 3 show that the GMM estimators revealed the same results as the OLS model. The impact of

equity capital was nonsignificant, while the impacts of assets management, assets quality, bank size and liquidity on banks' return were significant. All moderator effects observed in the OLS models were confirmed.

**Table 3.** Generalized method of moments model.

| | Model 1-ROE | Model 1-ROA | Model 2-ROE | Model 2-ROA |
|---|---|---|---|---|
| Liquidity Creation | −0.2293 *** | −0.2505 *** | −0.2513 *** | −0.2652 *** |
| Size | 0.0848 ** | 0.0955 ** | 0.0922 ** | 0.0923 |
| Equity Capital | 0.2842 | 0.2636 | 0.2433 | 0.2312 ** |
| Efficient Management of Assets | 0.9523 *** | 1.0935 *** | 1.0965 *** | 1.1134 |
| Asset Quality | −0.1352 ** | −0.1634 ** | −0.1549 * | −0.1622 * |
| Size*Liquidity Creation | | | −0.4322 *** | −0.4156 *** |
| Equity Capital*Liquidity Creation | | | −0.3055 | −0.3101 |
| Efficient Management of Assets*Liquidity Creation | | | −1.4086 | −1.1422 |
| Asset Quality*Liquidity Creation | | | −2.3077 *** | −2.5127 *** |
| Inflation | 0.0533 | 0.0537 | 0.0532 | 0.0544 |
| GDP | 0.1052 | 0.1045 | 0.1252 | 0.1384 |
| AR (1)-P | 0.0034 | 0.0601 | 0.0482 | 0.0247 |
| AR (2)-P | 0.3514 | 0.4059 | 0.6001 | 0.3491 |

\* $p < 0.05$, \*\* $p < 0.01$, \*\*\* $p < 0.001$, ROE = return on equity, ROA = return on assets.

The empirical findings in Table 4 reveal that the determinants of banks' return are similar regardless of the employed statistical model. The results showed that both the 2SLS model and the OLS regression had the same results and thus, indicate the absence of endogeneity problems that can bias the relationship between dependent and independent variables.

**Table 4.** Two-Stage least squares model.

| | Model 1-ROE | Model 1-ROA | Model 2-ROE | Model 2-ROA |
|---|---|---|---|---|
| Liquidity Creation | −0.2053 *** | −0.2505 *** | −0.2585 *** | −0.2644 *** |
| Size | 0.0835 ** | 0.0955 ** | 0.0955 ** | 0.0965 |
| Equity Capital | 0.28126 | 0.2636 | 0.2455 | 0.2366 ** |
| Efficient Management of Assets | 0.9855 *** | 1.0935 *** | 1.0966 *** | 1.1122 |
| Asset Quality | −0.1346 ** | −0.1634 ** | −0.1566 * | −0.1655 * |
| Size*Liquidity Creation | | | −0.4303 *** | −0.4155 *** |
| Equity Capital*Liquidity Creation | | | −0.3055 | −0.3144 |
| Efficient Management of Assets*Liquidity Creation | | | −1.4086 | −1.1442 |
| Asset Quality*Liquidity Creation | | | −2.3077 *** | −2.5952 *** |
| Inflation | 0.0535 | 0.0552 | 0.0544 | 0.0524 |
| GDP | 0.1054 | 0.1052 | 0.1295 | 0.1352 |
| $R^2$ | 0.5505 | 0.5066 | 0.5434 | 0.4847 |
| Adjusted $R^2$ | 0.5095 | 0.4569 | 0.5088 | 0.4359 |

\* $p < 0.05$, \*\* $p < 0.01$, \*\*\* $p < 0.001$, ROE = return on equity, ROA = return on assets.

## 5. Materials and Methods

### 5.1. Research Procedure and Samples

The research sample consisted of banks in the MENA region from 2020 to2021, which is the period over which the coronavirus was in the region. The data source was the Orbis Bankscope database. The data on the control variables was obtained from the World Bank database. Data was collected from 156 banks in 2020 and 142 banks in 2021, for a total of 298 banks during the entire 2020–2021 period. Banks from the 11 MENA countries of Egypt, Tunisia, Morocco, Qatar, Bahrain, Oman, Kuwait, Saudi Arabia, United Arab Emirates, Jordan, and Israel were included in the sample, as shown in Table 5. To eliminate bias, banks from countries experiencing non-COVID-19-related banking, political, and financial crises were eliminated.

**Table 5.** Research sample of MENA banks.

| Country | Banks 2020 | Banks 2021 | Total Number of Banks 2020–2021 |
|---|---|---|---|
| Egypt | 21 | 19 | 40 |
| Tunisia | 22 | 21 | 43 |
| Morocco | 15 | 13 | 28 |
| Qatar | 12 | 9 | 21 |
| Bahrain | 14 | 13 | 27 |
| Oman | 7 | 7 | 14 |
| Kuwait | 11 | 10 | 21 |
| Saudi Arabia | 12 | 12 | 24 |
| United Arab Emirates | 17 | 15 | 32 |
| Jordan | 13 | 12 | 25 |
| Israel | 12 | 11 | 23 |
| Total | 156 | 142 | 298 |

### 5.2. Research Instruments

The data was organized into a pooled time series covering the 2020–2021 COVID-19 lockdown period. Ordinary least squares regression was used, as observations failed to show any evidence of nonlinearity. Further perusal of the data showed no evidence of serial autocorrelation or heteroscedasticity, so that ordinary least squares regression could be used for data analysis. Data was entered in the SPSS (Statistical Package for the Social Sciences) program (Version 4) to generate two ordinary least squares regressions. The SPSS software has sufficient time series capabilities for ordinary least squares regressions. The impact on return on assets (ROA) and return on equity (ROE) (dependent variables) of the independent variables of liquidity (LIQ), size (SIZ), equity capital (CAP), the efficiency of asset management (ASE), and asset quality (ASQ), was measured in Model 1. The moderator effect of liquidity was determined by the products of liquidity with independent variables, including LIQ*SIZ, LIQ*ASE, LIQ*ASQ, and LIQ*CAP in Model 2. Control variables included Gross Domestic Product (GDP), and Inflation (INF). Equations (1) and (2) describe Model 1, while Equations (3) and (4) describe Model 2.

Model 1

$$ROA = \alpha + \beta_1 CAP + \beta_2 SIZ + \beta_3 ASE + \beta_4 ASQ + \beta_5 LIQ + \beta_6 INF + \beta_7 GDP + \varepsilon \tag{1}$$

$$ROE = \alpha + \beta_1 CAP + \beta_2 SIZ + \beta_3 ASE + \beta_4 ASQ + \beta_5 LIQ + \beta_6 INF + \beta_7 GDP + \varepsilon \tag{2}$$

Model 2

$$ROA = \alpha + \beta_1 CAP + \beta_2 SIZ + \beta_3 ASE + \beta_4 ASQ + \beta_5 LIQ$$
$$+ \beta_6 LIQ * CAP + \beta_7 LIQ * SI Z + \beta_8 LIQ * ASQ \quad (3)$$
$$+ \beta_9 LIQ * ASE + \beta_{10} INF + \beta_{11} GDP + \varepsilon$$

$$ROE = \alpha + \beta_1 CAP + \beta_2 SIZ + \beta_3 ASE + \beta_4 ASQ + \beta_5 LIQ$$
$$+ \beta_6 LIQ * CAP + \beta_7 LIQ * SI Z + \beta_8 LIQ * ASQ \quad (4)$$
$$+ \beta_9 LIQ * ASE + \beta_{10} INF + \beta_{11} GDP + \varepsilon$$

where,

- **Independent variables:**
  LIQ = liquidity creation = (total loans/total assets),
  SIZ = logarithm of total assets,
  CAP = equity capital = (total equity/total assets),
  ASE = efficiency of asset management = (total operating income/total assets)
  ASQ = asset quality = (total nonperforming loans/total loans)
- **Dependent variables:**
  ROA = return on assets = (net income/total assets)
  ROE = return on equity = (net income/equity)
- **Control variables:**
  GDP = Gross Domestic Product per country
  INF = inflation rate

A two-stage least squares model and generalized method of moments model evaluated the robustness of the ordinary least squares model. Robustness checks are essential to ensure that results do not contain spurious relationships. The two-stage least squares model eliminated the problem of possible endogeneity of independent variables.

## 6. Discussion and Conclusions

*6.1. Discussion of Key Findings*

6.1.1. Key Findings

The following key findings were obtained from the testing of hypotheses.
Main Effects

(1) The efficient management of assets significantly *increased* return on assets, and significantly *increased* return on equity.
(2) Bank size significantly *increased* return on assets, and significantly *increased* return on equity.
(3) Liquidity creation significantly *increased* return on assets, but significantly *decreased* return on equity.
(4) Asset quality significantly *increased* return on assets, but significantly *decreased* return on equity.

Moderator Effects

(1) Liquidity creation in conjunction with bank size significantly *increased* return on assets, but significantly *decreased* return on equity.
(2) Liquidity creation in conjunction with asset quality significantly *increased* return on assets, but significantly *decreased* return on equity.

6.1.2. Discussion

During the COVID-19 crisis, large banks drew on their reserves to maintain the deposits needed to make loans. Loan interest income remained at a stable level, resulting in increased return on assets. Stable loan interest incomes and stable net interest margins were supplemented by unchanged levels of shareholders' equity, as net income continued to be reinvested in the bank, maintaining the level of retained earnings. Return on equity

increased as well. The efficient management of assets also increased return on assets and return on equity. Bank managers who found additional revenue streams to offset deposit drains, were able to increase net income and net interest margins.

On the other hand, liquidity creation in the form of loan-to-asset ratio and asset quality, increased return on assets, while reducing return on equity. Mdaghri (2022) and Sahyouni and Wang (2019) noted that banks in the MENA region had high volumes of nonperforming loans. We may expect that the quantity of such nonperforming loans increased during the financial crisis. While the distribution of loans increased, resulting in higher net interest income, and in turn, higher return on assets, the ability of these loans to generate the retained earnings was limited by the presence of defaulted loans. This reduction in retained earnings reduced the amount of equity capital, reducing returns to shareholders or return on equity. Likewise, nonperforming loans reduced asset quality. The reduction in asset quality was viewed as a negative signal by shareholders, who may have sold their stock in the bank. It follows that the return to the shareholders decreased, or the return on equity decreased. Nonperforming loans may also have caused the joint effect of loan creation on bank size and asset quality, respectively, to have reduced return on assets and return on equity. The benefit of increased return on assets and return on equity is that large banks have dissipates if the additional defaults on the loans. Loan creation, while creating more loans, fails to benefit large banks as the additional loans are in default. This is seen in the interaction of loan creation with asset quality. The creation of additional loans of poor quality reduces profitability, with both reduced return on assets and return on equity.

In accordance with the overall thesis of this study, loan creation increased profitability during the COVID-19 crisis in the MENA region. Managers who found alternative revenue streams were able to increase return on assets and return on equity. This was the case with large banks as well. However, the deteriorating environment increased the volume of nonperforming loans. Loan creation of a large number of nonperforming loans was not beneficial in fulfilling the financial intermediation role of banks of transforming deposits into loans, thereby increasing loan income. This reduced return on equity. In fact, unproductive loan creation reversed the benefits of large bank size on return on assets. The joint effect of loan creation of nonperforming loans and asset quality was the most powerful negative influence of all relationships examined.

*6.2. Theoretical Implications*

Efficient management of assets was found to increase return on assets and return on equity. This result is in keeping with Liu and Wilson (2010), who found that non-MENA banks purchased investment grade bonds and government securities during periods of uncertainty to maintain stable income levels. Similar revenue streams were sought in the pre-COVID-19 MENA region (El-Chaarani and Shaker 2018). Our study extends these findings to crisis periods, so that bank profits may be maintained with judicious investments in derivatives, loan commitments, and letters of credit.

Regarding bank size, El-Chaarani et al. (2022) found that during the crisis, the use of fintech and big data increased customer satisfaction in large banks. We may extend this finding to the crisis period, as such tools may have resulted in large banks increasing return on assets and return on equity.

Although Al-Matari (2021) observed the moderation of the equity capital–profitability relationship into MENA banks before the crisis, we found no such practice during the COVID-19 lockdown period. We partly supported his finding of liquidity creation improving bank performance, measured by return on assets. However, we found that liquidity creation reduced return on equity, presumably due to nonperforming loans.

Successive studies reported the presence of nonperforming loans in MENA banks in the pre-crisis period (Saif-Alyousf et al. 2017; Kadioglu et al. 2017). Bloem and Gorter (2001) found that such loans caused depositors to withdraw their savings from banks. The ability of nonperforming loans to send strong negative signals to bank performance in our

study was underscored by loan creation reducing the positive effects of bank size and loan quality on return on assets when measured individually, to negative effects on return on equity when measured in conjunction with loan creation.

This study provides new insights to theories that analyze the financial behavior and capital structure of firms and their impact on the financial return during unstable periods. First, this paper validates the marketing timing theory that proposes the adjustment of financial behavior and capital structure of banks to optimize the level of financial return. It also supports the income anticipation theory that assumes that banks must secure the liquidity level by the liquidation of long-term loans. However, the pecking order theory and agency theory are not supported since the level of equity capital did not influence bank profitability during the crisis. Second, this paper delivers to scholars in the finance and banking sector a global comprehension of key success factors that determine the financial return of banks during crisis across 11 countries.

### 6.3. Practical Implications

The study has several practical implications. First, MENA banks must prioritize the reduction of nonperforming loans, as it is these loans that negate increases in profits for banks of large size and asset quality. Loan approvals must be more discerning, with borrowers with weak creditworthiness being denied. Second, liquidity creation must be pursued during noncrisis periods, with management embarking on an outreach program towards large depositors, institutional depositors, and foreigners to encourage them to increase their deposits at the banks. Third, during crises, effective management of liquidity creation is essential. As is the case with Saudi banks, other banks may pursue loan commitments, letters of credit, and derivatives, all of which provide alternate revenue streams to loan interest income. In other words, diversification of revenues provides banks with multiple income streams, when there are restrictions on the main source of income. This recommendation suggests the need to hire skilled investment professionals, who are knowledgeable about the sophisticated financial instruments. Fourth, as Mdaghri (2022) revealed, MENA banks finance economic growth by reducing the external funding cost for government. Banks engaged in financing growth may work in conjunction with international aid organizations that provide funding. Joint MENA bank and international aid loans will reduce the cost of funding to government borrowers, thereby benefitting society. Beck et al. (2013) and Bayar (2019) have shown that the defaults on loans issued for developmental purposes in the MENA region had lower rates of default than typical loans.

Two policy recommendations emerge from this study. As large banks have the advantage of large deposits and large reserves, small banks must consider being acquired, or merging with large banks. The resulting financial institution will be sufficiently large to be able to maintain the inflow of deposits to create loans, even during a banking crisis. The dependence on banking by businesses and government in the MENA region is due to underdevelopment of capital markets. As a long-term goal, MENA governments should create securities markets of stocks, bonds, mutual funds, and exchange-traded funds. An active securities market will permit funds to raise capital from the sale of stocks and bonds. Cash management accounts and money market funds may provide the cash needed by businesses, in lieu of bank loans. In essence, banks will be less affected by crises, if the dependence on them for funding both businesses and government is partly relieved by financial markets.

### 6.4. Conclusions

This study fulfilled its purpose of explaining the role of liquidity creation, or bank loan formation, in the MENA region during the COVID-19 crisis. Liquidity creation, as a main effect, influenced bank profitability, as did bank size, asset quality, and efficient management of assets. We recommended that banks manage assets efficiently to maintain deposit levels during crises. We also noted the importance of bank size and high asset

quality in loan creation. However, simply creating loans through inflows of deposits is insufficient. The loans must be performing loans that generate a stream of interest income. This conclusion was underscored by the finding that the joint effects of liquidity creation and bank size and liquidity creation and asset quality reduced return on equity. We feel that that the presence of nonperforming loans negated the positive effect of bank size and asset quality on return on equity. Future research must empirically explore the influence of nonperforming loans on bank liquidity, return on assets, and return on liquidity.

We found that liquidity creation and asset quality, as main effects, increased return on assets, but reduced return on equity. Why were there differential effects on the two measures of profitability? Return on assets is less dependent on the quality of loans. It is the return on *all* loans, performing and nonperforming, and does not distinguish between them. Return on equity includes the addition to retained earnings from interest income, which is obtained only from performing loans. Therefore, return on equity captures the reduction in retained earnings from nonperforming loans.

## 7. Research Limitations

This research has several limitations that could be resolved in future studies. First, this paper is confined to a very short period of only two years. Therefore, it is suggested that the period of this study be extended by considering the pre- and post-pandemic periods. Second, this research does not account for inter-country differentials in bank performance. In future research, this problem could be resolved by studying the banking sector in each country. Third, this paper considers inflation, and Gross Domestic Product as control variables, while omitting other factors that could influence banks, such as legal protection and oil prices. Finally, the number of banks considered in this paper is 156, which is a relatively small sample, since it is extracted from 11 countries. This problem could be resolved in future research be extending the number of banks to be examined.

**Author Contributions:** Conceptualization: H.E.-C. and G.A., methodology: H.E.-C. and G.A., validation: H.E.-C. and R.A., formal analysis: H.E.-C., investigation: H.E.-C., resources: G.A., data curation: H.E.-C., writing: original draft preparation: R.A., writing: review and editing, H.E.-C. and R.A., visualization: H.E.-C., supervision: H.E.-C., project administration: R.A. All authors have read and agreed to the published version of the manuscript.

**Funding:** This research received no external funding.

**Data Availability Statement:** Data is available from the first author upon request.

**Conflicts of Interest:** The authors declare no conflicts of interest.

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
