# Peer review of "The Role of Liquidity Creation in Managing the COVID-19 Banking Crisis in Selected Mena Countries"

_ijfs, doi:10.3390/ijfs11010039_

Round 1

Reviewer 1 Report

I have now completed my assessment of the manuscript titled “DETERMINANTS OF BANK FINANCIAL PERFORMANCE IN THE MENA REGION DURING THE COVID-19 LOCKDOWN: DID LIQUIDITY MATTER?”. However, following are some of my comments which I feel the authors must address before it can be accepted for publication in the IJERPH journal:

1. The title should attract the audience.

2. The abstract should state in some detailed form for the purpose of the research, the principal results, and major conclusions.

3. The current study lacks motivation. Authors should expand the main goals and objectives of this work to improve on the motivation of the current study. The novelty of this work is not clear.

4. I would like to suggest that authors should update the literature part. Specifically, the latest research trends, and in order to highlight the academic frontier of the research, the references of the recent year need to be referenced. 

5. I'll advise authors to separate the "Literature review and Hypothesis development" section into its own section.

6. More importantly, per my critical observation, the authors in this manuscript reported results from the analysis without discussing them. I, therefore, suggest that authors should instead separate the presentation of results from the discussions. Both should be in different sub-sections under the title "Results and discussions. Moreover, the authors are supposed to give the impact mechanism behind the results obtained (that is, detailed economic meaning of their results and implications or, in other words, what might have brought about the results they have obtained). It must be done systematically by comparing them to further studies in the related fields.

7. The policy recommendation is too slim. Authors should add more to this section, especially in the aspect of policy framing and implementation.

8. Method seems fine to me.

9. There are several grammatical issues throughout the paper. Please ask a professional copy-editing company for the English editing.

10. Modify the paper based on the journal’s guidelines.

Author Response

See attached file, for table for Reviewer 1. Changes in the document are in yellow.

Reviewer 2 Report

Thank you for giving me an opportunity to review this paper: Determinants of Bank Financial Performance in the MENA Region During the Covid-19 Lockdown: Did Liquidity Matter?

There are several suggestions given for the improvement of the paper:

1. Originality: Does the paper contain new and significant information adequate to justify publication?

1.     Overall, the introduction is clearly written. However, it is too lengthy. Thus, I suggest author to trim it down to maximum three pages.

2.     What is the gap of the study? What is the aim of the study? Perhaps author should make the gap clearer and relate it back to the issue. The issue is overwhelmingly explained. It should be streamline further and talk about the gap.

3.     Further to this, authors should also present several past studies in the introduction. From there, authors may be able to shows the uniqueness of the study. How your study stands uniquely as compared with others?

4.     What is the expected contribution in this study? This should be stated in the introduction.

5.     The figure 1 is not clear. Please redraw the figure.

6.     The methodology is not well written. Should provide more detail. It should be reorganized into two sub sections: (1) research procedure and samples and (2) research instruments.

7.     What software used in this study? Authors are required to provide justification.

8.     The discussion and conclusion section structure should be revised as follows:

Discussion of key findings

Theoretical Implications

Practical/Managerial Implications

Limitations and Future Research

Conclusion

9.     For discussion, authors need to ensure the key findings are discussed. The current version of the discussion is poorly written which need to be strengthened. The discussion section is where you delve into the meaning, importance and relevance of your results. It should focus on explaining and evaluating what you found, showing how it relates to your literature review and research questions, and making an argument in support of your overall conclusion.

10.  Should have a standalone section for theoretical implication. The current form of theoretical implication is too shallow. How do you imply these findings and compared with past study’s findings?

11.  Should have a standalone implication. The current form of practical implication is too shallow. I would suggest author to provide implications based on the current practices and policies.

12.  Conclusion is required to ensure it is reflecting the introduction and objective of the study.

Author Response

Changes in the document were made in purple. See the attached point by point list.

Reviewer 3 Report

The paper is not respecting the journal template! The figure and tables must be made much more professional. 

Please specify when the first time used, what is the meaning of MENA region (abstract and introduction). 

The authors state in the abstract that the observations were made after collecting data from 11 countries. So not all countries from the MENA region were included in the study? Is it correct? Please revise the title and the text if so. 

I do not find necessary the last paragraph from the introduction. Instead, the authors should state there the aim of the paper. 

The last paragraph from the subsection "Theories of Liquidity" needs to be sustained by at least one citation. If the statements are the author's conclusion, please rephrase them. 

The first sentence from " Liquidity Creation" - based on what the authors give this definition?

The first half of the second paragraph from the " Liquidity Creation" subsection needs more citations. 

Please explain any abbreviation the first time it appears in the text (i.e., OLS page 17). 

Please state the scientific criteria determining the author's use of the described methodology. 

Tables and figures should be stand-alone! Please add as a footnote of the tables all the abbreviations used in the tables. 

Please make "Research limitations" as a new chapter (7.).

The most important part of the paper is missing! The "Discussions" or "Results and discussion"  section should be added! 

I appreciate how the authors discuss the literature in the introduction. The same thing should be done concerning the obtained results. 

Author Response

Changes to the document were made in green. See table attached with list of changes for Reviewer 2 in green. 

Round 2

Reviewer 2 Report

All comments are addressed by authors. Thus, I have no further comments.

Author Response

No changes requested.

Reviewer 3 Report

The changes are hard to follow because the paper is not in the journal template and has no lines to address!

Overall, the paper is easier to follow. 

Were is the figure? Is stil mentioned in the text. 

Author Response

The reference to Figure 1 has been removed from page 11.
